# Study on the Influence of Runner and Overflow Area Design on Flow–Fiber Coupling in a Multi-Cavity System

**DOI:** 10.3390/polym16091279

**Published:** 2024-05-02

**Authors:** Fang-Lin Hsieh, Chuan-Tsen Chen, Shyh-Shin Hwang, Sheng-Jye Hwang, Po-Wei Huang, Hsin-Shu Peng, Ming-Yuan Jien, Chao-Tsai Huang

**Affiliations:** 1Department of Chemical and Materials Engineering, Tamkang University, New Taipei City 251301, Taiwan; dick64049@gmail.com (F.-L.H.); a9500521@gmail.com (C.-T.C.); 2Department of Mechanical Engineering, Chien Hsin University of Science and Technology, Taoyuan City 320678, Taiwan; stanhwang@uch.edu.tw (S.-S.H.); jackyaren@uch.edu.tw (M.-Y.J.); 3Department of Mechanical Engineering, National Cheng Kung University of Engineering, Tainan 701401, Taiwan; jimppl@mail.ncku.edu.tw; 4Department of Mechanical and Computer Aided Engineering, Feng Chia University of Engineering and Science, Taichung 407102, Taiwan; bowei8915@gmail.com (P.-W.H.); hspeng@fcu.edu.tw (H.-S.P.)

**Keywords:** injection molding, fiber-reinforced thermoplastics (FRPs), fiber orientation, flow–fiber coupling, a multi-cavity system, overflow area

## Abstract

Fiber-reinforced composites (FRPs) are characterized by their lightweight nature and superior mechanical characteristics, rendering them extensively utilized across various sectors such as aerospace and automotive industries. Nevertheless, the precise mechanisms governing the interaction between the fibers present in FRPs and the polymer melt during industrial processing, particularly the manipulation of the flow–fiber coupling effect, remain incompletely elucidated. Hence, this study introduces a geometrically symmetrical 1 × 4 multi-cavity mold system, where each cavity conforms to the ASTM D638 Type V standard specimen. The research utilizes theoretical simulation analysis and experimental validation to investigate the influence of runner and overflow design on the flow–fiber coupling effect. The findings indicate that the polymer melt, directed by a geometrically symmetrical runner, results in consistent fiber orientation within each mold cavity. Furthermore, in the context of simulation analysis, the inclusion of the flow–fiber coupling effect within the system results in elevated sprue pressure levels and an expanded core layer region in comparison to systems lacking this coupling effect. This observation aligns well with the existing literature on the subject. Moreover, analysis of fiber orientation in different flow field areas reveals that the addition of an overflow area alters the flow field, leading to a significant delay in the flow–fiber coupling effect. To demonstrate the impact of overflow area design on the flow–fiber effect, the integration of fiber orientation distribution analysis highlights a transformation in fiber arrangement from the flow direction to cross-flow and thickness directions near the end-of-fill region in the injected part. Additionally, examination of the geometric dimensions of the injected part reveals asymmetrical geometric shrinkage between upstream and downstream areas in the end-of-fill region, consistent with microscopic fiber orientation changes influenced by the delayed flow–fiber coupling effect guided by the overflow area. In brief, the introduction of the overflow area extends the duration in which the polymer melt exerts control in the flow direction, consequently prolonging the period in which the fiber orientation governs in the flow direction (A_11_). This leads to the impact of fiber orientation on the flow of the polymer melt, with the flow reciprocally affecting the fibers. Subsequently, the interaction between these two elements persists until a state of equilibrium is achieved, known as the flow–fiber coupling effect, which is delayed.

## 1. Introduction

In recent times, fiber-reinforced thermoplastics (FRPs) have become increasingly popular for creating lightweight vehicle body parts [1,2,3] and in renewable energy equipment [4,5] manufacturing due to their exceptional mechanical properties. Various researchers have delved into this field to understand how fibers enhance plastic properties. Thomason [6,7] introduced empirical guidelines to help the industry determine the minimum fiber length required for auto parts to possess adequate rigidity and impact strength. Other scholars [8,9] have explored ways to enhance product mechanical properties by combining different materials and formulas. Additionally, many studies [10,11,12,13,14,15,16] have focused on the impact of process-induced fiber microstructural changes. Rohde et al. [10] used thermal degradation to analyze how process parameters affect fiber length variation, while Goris et al. [11] developed the PEC technique to attribute fiber length post-injection molding, identifying screw speed and back pressure as crucial factors in fiber breakage. However, research [11,12] has shown that measuring small fiber amounts in each sample can lead to significant inaccuracies, necessitating larger experimental datasets for reliable results in fiber breakage studies, which poses a significant challenge.

Moreover, to enhance comprehension of the alterations in microstructural characteristics of fibers during the injection molding process, Folgar and Tucker [13] and Advani and Tucker [14] introduced theoretical models aimed at predicting the fiber orientation of short fibers in FRP injection molding. These models have proven to be valuable tools for guiding the design and advancement of FRP injection products in contemporary industry and academia [15]. However, Thomason [6,7] emphasized the importance of retaining a sufficient length of fibers in the final product to meet impact strength requirements in practical applications. The challenge lies in effectively incorporating long fibers into the polymer matrix and preserving their length post-injection, a task that poses difficulties whether approached through theoretical predictions or experimental observations. Scholars have subsequently put forth the Anisotropic Rotary Diffusion (ARD) model [16] and the Reduced Strain Closure (RSC) model [17,18] to delve deeper into the microstructural properties of long fiber-reinforced plastics. These numerical theoretical models, albeit beneficial, necessitate numerous additional parameters and are intricate to apply. Subsequently, Tseng et al. [19] simplified the model parameters with the iARD-RPR model, which was integrated into the commercial software Moldex3D 2022.

Furthermore, during the injection molding process, the flow of molten melt significantly impacts fiber orientation, while the presence of fibers can also influence the flow of the polymer melt. This reciprocal interaction between melt flow and fibers is referred to as the flow–fiber coupling effect. Numerous researchers have endeavored to investigate these effects, as evidenced by previous studies [20,21,22,23,24,25]. Notably, Lipscomb et al. [20] introduced Lipscomb’s constitutive equation to analyze the flow–fiber coupling effect in an axisymmetric contraction (4.5:1) flow system. Their findings indicated that a higher flow–fiber coupling leads to the generation of a larger corner vortex. Subsequently, VerWeyst and Tucker [21] further explored this effect using the same flow system and introduced the concept of Np (particle number) to quantify the coupling effect, corroborating Lipscomb’s results. They integrated flow stress and fiber orientation, employing 2.5D flow simulation to investigate the flow–fiber coupling effect in a center-gated disk for injection molding across various Np values. Their analysis revealed that an increase in Np (indicating higher flow–fiber coupling) resulted in minimal changes in streamlines and velocity profiles, with fiber orientation tensor components showing little dependence on Np values. Additionally, Tseng and Su [22] expanded upon the concepts introduced by Lipscomb, VerWeyst, and Tucker, incorporating them into a 3D flow simulation. The study revealed that increasing the Np number did not significantly alter the velocity profiles and fiber orientation tensor components. However, it was noted that when Np reached a high value (Np = 20), the computation became divergent. Criticisms were raised regarding the inadequacy of the Lipscomb constitutive equation in predicting fiber orientation in injection molding simulations. To address these issues identified in previous studies [20,21,22], Favaloro et al. [23] introduced the informed isotropic viscosity model (referred to as the IISO viscosity model). This model was developed to account for variations in the stress tensor based on the deformation mode relative to the orientation state. The researchers observed a plug flow behavior in velocity profiles and obtained the border core width of the fiber orientation tensor in the flow direction (A_11_) under the influence of flow–fiber coupling in a center-gated disk for injection molding, contrasting the findings of VerWeyst and Tucker [21]. Subsequently, Tseng and Favaloro [24] refined the IISO model into the revised IISO model and applied it to investigate the flow–fiber coupling effect in an axisymmetric contraction flow system and an end-gated plate, noting a transition in flow front behavior from convex–flat–flat to convex–flat–concave with coupled flow and fiber interactions. The researchers highlighted similarities between the border core width of the fiber orientation tensor in the flow direction (A_11_) and the impact of yield stress viscosity at low shear rates on the flow field. Despite these advancements, limited practical verification has been conducted on flow–fiber coupling effects. Huang and Lai [25] expanded on the work of Tseng and Favaloro [24] by examining a more complex center-gated plate and confirmed the presence of convex–flat–concave flow front behavior when considering flow–fiber coupling effects. They further demonstrated that the flow–fiber coupling effect was prominent at the end of the filling region (EFR) but not in the near-gate region (NGR) through numerical simulations and experimental observations, attributing the diminished effect in the NGR to strong shear forces.

In addition, some researchers [26,27,28] have used traditional optical and tomographic techniques to examine the fiber microstructures in injected FRP components. Bernasconi et al. [26] discovered that the classical optical section method may not always provide accurate fiber orientation measurements and emphasized the importance of selecting the correct section plane. They noted that when the appropriate section plane is chosen, the classical optical section method (which is fully destructive) can yield results similar to the nondestructive micro-CT method. Gandhi et al. [27] also employed the micro-CT method to capture three-dimensional images of fibers and analyze their orientation distribution, mentioning the challenge of handling large image data. By implementing specific image technologies to reduce data size, they claimed that the micro-CT approach can produce results comparable to the conventional ellipse method. Additionally, Teuwsen et al. [28] utilized micro-CT scan analysis and VG Studio image analysis to investigate fiber orientation and concentration distribution, revealing nonuniform patterns along the part thickness and flow path. They identified a distinct seven-layered fiber orientation pattern through the thickness direction of the injected part. Huang et al. [29] used micro-CT scans and Avizo image analysis to explore the relationship between fiber orientation and tensile stress using three different standard tensile bar systems. They showed that the design of the side gate can enhance fiber orientation distribution in the flow direction, leading to increased mechanical strength. Based on the literature, many researchers believe that the superior performance of FRPs compared to general plastics is primarily attributed to their fiber microstructural properties, such as fiber orientation, length, and concentration distribution. They also highlighted that the mechanical properties of these components heavily rely on fiber orientation and length. Despite researchers employing various techniques, both nondestructive and destructive, understanding the fiber reinforcement mechanism through experimental methods remains challenging. Later, Huang and Lai [25] utilized theoretical simulation analysis, micro-computed tomography, and image processing technology to investigate the interaction between flow–fiber coupling. Their research revealed that at the end of the filling process, the decrease in shear stress in the polymer melt area led to a pronounced coupling effect between flow and fiber orientation. Subsequently, Huang et al. [30] conducted a study on the fiber orientation variation, considering the flow–fiber coupling effect. They also examined the geometric shrinkage characteristics of the product in different directions. The research demonstrated that the coupling effect between flow and fiber orientation caused a shift of the fiber orientation tensor in the flow direction (A_11_) towards the cross-flow (A_22_) and the thickness (A_33_) directions, resulting in consistent resistance to shrinkage in these directions. Consequently, the ability to effectively and significantly modify the flow field to specifically control changes in fiber orientation within the polymer melt is limited.

In contrast, while polymer melt is commonly conveyed through conventional runners, it lacks effective adjustability within the mold cavity. However, the existing literature indicates that alterations in the flow path process can potentially lead to a reversal of the polymer melt, thereby modifying the characteristics of the melt flow field and subsequently impacting quality. Notably, Beaumont and colleagues [31,32,33] have extensively investigated this phenomenon in contemporary times, culminating in the development of a commercial product known as MeltFlipper^®^. Initially, when the polymer melt traversed through geometrically balanced runner systems, the resulting finished product differed from the anticipated outcome. The researchers attributed this discrepancy to the inherent generation of shear heat during melt flow, prompting them to introduce a melt flip design element into the flow channel to counteract the unbalanced polymer melt flow. Subsequent studies by Chien et al. [34,35] employed the finite volume method to simulate polymer melt behavior in a cold runner system, elucidating that flow imbalance arises from shear-induced heat generation, leading to temperature gradients and viscosity alterations within the material. This imbalance ultimately impacts product quality. Additionally, they demonstrated that melt flipping induces a rearrangement of the flow field, further influencing melt flow. Recently, Petzold [36] proposed optimizing the temperature field of hot runner systems, while Krzysztof [37] introduced strategies to address filling imbalances in geometrically balanced mold systems, highlighting the efficacy of the ANN method as an optimizer. Collectively, these studies offer valuable insights into the potential for altering the intrinsic quality of the melt through modifications in the flow channel within injection molding systems.

Based on the above information, this study examines how altering the flow field can affect fiber orientation and flow–fiber coupling. To explore these effects, we devised a geometrically symmetrical 1 × 4 multi-cavity mold system with overflow areas at the end of each cavity. By implementing a geometrically symmetrical runner system, we aimed to manipulate fiber orientation throughout the mold. The addition of overflow areas at the end of the filling zone was intended to assess any changes in the flow–fiber coupling effect. Further elaboration on these processes and outcomes will be provided in subsequent sections.

## 2. Theoretical Background

### 2.1. Model for Polymer Melt Flow

In the course of injection molding, polymeric materials, whether containing fibers or not, are characterized as compressible, non-Newtonian fluids. The governing equations for fluid mechanics in this context pertain to three-dimensional transient nonisothermal motion [19]:(1)$$\frac{\partial \rho }{\partial t}+\nabla \;\cdot \rho u=0,$$
(2)$$\frac{\partial }{\partial t}\left(\rho u\right)+\nabla \cdot \left(\rho uu\right)=\nabla \cdot \sigma +\rho g,$$
(3)σ=−PI+τ,
(4)$$\rho C_{P}\left(\frac{\partial T}{\partial t}+u\cdot \nabla T\right)=\nabla \cdot \left(k\nabla T\right)+\tau :D,$$
where ρ is density; t is time; **u** is the velocity vector; **σ** is the total stress tensor; **g** is the acceleration vector of gravity; **τ** is the extra stress tensor; *P* is pressure; C_P_ is specific heat; T is temperature; k is thermal conductivity; and **D** is the rate-of-deformation tensor.

Furthermore, in the context of a polymer melt, the extra stress tensor can be illustrated as follows:(5)τ=2ηD,
where η is the shear viscosity of a polymer melt.

In order to analyze the temperature-dependent viscosity of a polymer melt, the modified Cross model is employed in the following manner:(6)$$\eta \left(T,\dot{\gamma }\right)=\frac{\eta_{o}\left(T\right)}{1+{\left(\eta_{o}\dot{\gamma }/\tau^{*}\right)}^{1-n}},$$
(7)$$\eta_{o}\left(T\right)=\mathrm{BExp}\frac{T_{b}}{T},$$
where η_0_ is the zero shear viscosity; n is the power law index; and τ* is the parameter that describes the transition region between the zero shear rate and the power law region of the viscosity curve.

### 2.2. Models for Fiber Orientations

In the injection molding of FRP materials, individual fibers within the polymeric matrix are considered as axisymmetric rigid bonds. The orientation of each rigid bond is represented by the unit vector p along its axis direction, denoted as the fiber orientation. The collective orientation of a group of fibers is characterized by a second-order orientation tensor:(8)$$A=\oint \psi \left(p\right)pp\;dp,$$
where ψ(p) is the probability density distribution function over orientation space. Tensor **A**_4_ is a fourth-order orientation tensor, defined as follows:(9)$$A_{4}=\oint \psi \left(p\right)pppp\;dp.$$

To handle this complicated tensor system, Tseng et al. [19] extended ARD-RSC models [16,17,18] to develop a fiber orientation model to couple with Jeffery’s hydrodynamic (HD) model, namely, the iARD-RPR model:(10)$$\dot{A}={\dot{A}}_{HD}+{\dot{A}}_{\mathrm{iARD}}\left(C_{I},\;C_{M}\right)+{\dot{A}}_{\mathrm{RPR}}\left(\alpha \right),$$
where $\dot{A}$ represents the material derivative of A. Parameters C_I_ and C_M_ describe the fiber–fiber interaction and fiber–matrix interaction, while parameter α can slow down the response of fiber orientation. Jeffery’s hydrodynamic (HD) term can be written as follows:(11)$${\dot{A}}_{\mathrm{HD}}=\left(W\cdot A-A\cdot W\right)+\xi \left(D\cdot A+A\cdot D-2A_{4}:D\right),$$
(12)$$D=\frac{1}{2}(\nabla u+\nabla u^{T})$$
(13)$$W=\frac{1}{2}(\nabla u-\nabla u^{T})$$
where **W** and **D** are the vorticity tensor and the rate-of-deformation tensor, respectively. ξ is a shape factor of a particle. The rest of the details for the RPR model and the iARD model are available elsewhere [19].

### 2.3. Model for Viscosity by Flow–Fiber Coupling

In order to assess the intricate interaction between the flow and fiber, the updated IISO constitutive equation is employed. The outlined methodology for solving this issue is as follows. Initially, the governing equations governing the flow field and fiber orientation are calculated using a 3D finite volume approach. Subsequently, the velocity field and fiber orientation field are derived and utilized to ascertain the revised IISO viscosity. This iterative process continues until convergence is achieved. Further elaboration on this procedure can be found in the relevant literature [24]. Specifically, the modified IISO viscosity is expressed as detailed below:(14)$$\eta^{IIOS}=\left(1+R_{T}K_{s\;}\right)\eta_{S},$$
(15)$$R_{T}\left(\dot{\gamma }\right)=\frac{R_{T}^{0}}{1+{\left(\dot{\frac{\dot{\gamma }}{{\dot{\gamma }}_{C}}}\right)}^{2}},$$
(16)$$K_{s}=\frac{D:A_{4}:D}{2D:D}$$
where $\eta_{S}$ the nonlinear Newtonian viscosity for the fiber-filled polymer fluids and is described by the modified Cross model; *R_T_* is the dimensionless Trouton ratio parameter as a function of the strain rate; $R_{T}^{0}$ is the initial value of *R_T_*; *K_S_* is a stretching kernel that is related to the flow fields and the fiber orientation state; and ${\dot{\gamma }}_{C}$ is the critical strain rate (1/s).

## 3. Systems and Information

### 3.1. System of Simulation

To study the influence of the runner and overflow area design on the flow–fiber coupling, the geometrical configuration of the previous study [30] has been adopted for comparison purposes, as shown in Figure 1a. Specifically, it is a plate system with three different ASTM D638 standard specimens. As pointed out in the red color for the melt entrance, it is a runnerless system. Moreover, the runner and cavity configurations in this study are depicted in Figure 1b. Four models of identical size (referred to as S1 to S4) conforming to ASTM D638 standards were fabricated within the mold, each with a dimension of 63.5 mm × 9.53 mm × 3.5 mm. To investigate variations in fiber orientation, specific locations were chosen for observation, as illustrated in Figure 1c. The detailed structure and dimensions are exhibited in Figure 1d. Notably, Points B1 to B5 correspond to the near-gate region (hereafter referred to as the NGR), and Points H1 to H5 signify the end of the filling region (hereafter referred to as the EFR). Additionally, Figure 2 displays the layout of the mold base and cooling channels. To obtain data on fiber orientation distribution, the S1 sample is employed as an example. The sample can be distinguished into the near-gate region (NGR) and the end of the filling region (EFR), as shown in Figure 1e, for further obtaining the fiber orientation data in those certain measuring locations. The procedure involves pre-cutting 11 measuring nodes in the thickness direction at the designated measurement position, followed by the computation of fiber orientation for each measuring node in this direction using simulation analysis. The outcomes of this analysis are depicted in the Section 4 Results. It is important to note that the methodology for the remaining measurement positions (B1 to B5 and H1 to H5) remains the aforementioned approach.

Furthermore, in order to investigate the influence of design on the variation in flow–fiber coupling, the operational parameters include a filling time of 0.1 s, packing time of 5 s, packing pressure of 120 MPa, cooling time of 30 s, melt temperature of 230 °C, and mold temperature of 25 °C. Moreover, the materials utilized in this research consist of pure polypropylene (referred to as PP) and polypropylene containing 30% short fibers (referred to as 30SFPP). The specific grade names for these materials are Globalene ST868M for PP and Globalene SF7351 for 30SFPP, both of which are commercially available and supplied by LCY Chemical in Tainan City, Taiwan.

### 3.2. System of Experimentation

Figure 3a displays the Arburg 420C 1000-350-40 injection machine system provided by Arburg Co., Lossburg, Germany. The actual mold structure is depicted in Figure 3b. The materials utilized and the process conditions adopted are the same as those employed in the simulation system. To assess the impact of the flow–fiber coupling effect on the predicted fiber orientation distribution from numerical simulation, micro-computerized tomography (μ-CT) scanning was conducted on the tensile specimens. This scanning was carried out using a Bruker Skyscan 2211 with 40–190 kV and a resolution of 5 μm, facilitated by the MCL Multiscale X-ray CT laboratory at the Industrial Technology Research Institute in Taiwan. Subsequently, the scanned images were processed and analyzed utilizing Avizo software 2022 version. The details are available elsewhere [30].

### 3.3. Dimensional Measurement of the Injected Parts

In the context of geometric size measurement techniques and data processing for injected parts, taking the near-gate region (NGR) as an example, the procedure involves defining various sides based on different perspectives. For instance, the two sides in the flow direction are denoted as (Lx)_U_ and (Lx)_D_ from the front view perspective, while the sides in the cross-flow direction are labeled as (Ly)_L_ and (Ly)_R_. Additionally, the sides in the thickness direction are determined using a side view perspective as (Lz)_L_ and (Lz)_R_, as illustrated in Figure 4a–c. The straightforward geometry of the sample allows for precise measurement using a vernier ruler on all six sides. To ensure the accuracy and representativeness of the measurements, three samples are randomly selected for measurement, and the averages are calculated along with their standard errors. The detailed outcomes are presented in Table 1. Given the uncomplicated geometry of the final product and the precise measurement technique employed, the likelihood of errors is minimal. In other areas and simulation analysis, the approach is the same. Clearly, the standard deviation is about 0.000 to 0.01 mm.

## 4. Results

### 4.1. Flow Behavior Validation

To investigate the interaction between the flow and fiber in composite materials, it is essential to employ both theoretical analysis and experimental validation. The first stage of this study involves confirming the controllability of flow field characteristics. The investigation approach utilizes a filling volume of 76% as an example. Initially, computer-aided engineering (CAE) simulation is employed to capture the melt flow front outcomes when the injection filling volume reaches 76%, as depicted in Figure 5a. Subsequently, the actual filling volume was adjusted to 76% for a real injection molding process. The resulting actual melt flow front behavior is illustrated in Figure 5b. Subsequent to this, simulation analysis was conducted to compare the actual injection melt flow front outcomes. It was observed that the flow characteristics exhibited a high degree of consistency. Further comprehensive exploration of flow behavior is recommended for a more thorough understanding. As depicted in Figure 5c, it is observed that the polymer melt enters the NGR when the filling ratio reaches 60%. Subsequently, the filling proportion gradually increases from 76% to 84%, with the polymer melt transitioning into the EFR through the central region (referred to as CR), ultimately filling the mold cavity. Through computational analysis using CAE simulation and experimental observations, it is evident that the actual flow behavior is consistent with the simulated results.

### 4.2. Fiber Orientation Distribution from Cavity to Cavity

#### 4.2.1. Simulation Result with Flow–Fiber Coupling Effect

In the context of a four-cavity system with geometric symmetry, this study delves into the alterations in fiber orientation across various mold cavities within the NGR, while accounting for flow–fiber interactions. The fiber orientation tensor along the flow direction (A_11_) in the four mold cavities is depicted in Figure 6a, illustrating a gradual increase from 0.7 at the boundary to 0.85 at the shear layer, followed by a decrease to 0.35 at the core layer. Notably, the results obtained from mold cavities S1 to S4 exhibit a high degree of similarity. Additionally, Figure 6b,c present the fiber orientation tensor along the cross-flow direction (A_22_) and the thickness direction (A_33_), respectively, revealing comparable outcomes across mold cavities S1 to S4. In summary, the investigation indicates that, within the NGR, the fiber orientation remains consistent across different mold cavities in a geometrically symmetrical one-by-four cavity system.

Further investigation is needed to examine the alterations in fiber orientation within various mold cavities while taking into account the flow–fiber interaction in the EFR. In Figure 7a, the fiber orientation tensor along the flow direction (A_11_) is depicted at the end of the filling region for four cavities. The orientation gradually increases from 0.65 at the boundary to 0.85 at the shear layer, then decreases to 0.4 at the core layer. Notably, the results from mold cavities S1 to S4 exhibit a high degree of similarity. Additionally, Figure 7b,c illustrate the fiber orientation tensor along the cross-flow direction (A_22_) and the thickness direction (A_33_), respectively. It is evident that the outcomes in mold cavities S1 to S4 are nearly identical. In summary, in the context of a geometrically symmetrical one-by-four cavity system within the EFR, the fiber orientation remains consistent across different mold cavities.

#### 4.2.2. Simulation Result without Flow–Fiber Coupling Effect

Furthermore, in the absence of flow–fiber coupling, what will happen to the fiber orientation changes in different mold cavities? Focusing on the fiber orientation distribution within different mold cavities in the NGR, the fiber orientation tensor at the flow direction (A_11_) in the four mold cavities was examined in Figure 8a, revealing a gradual increase from 0.7 at the boundary to 0.85 at the shear layer, followed by a decrease to 0.3 at the core layer. Notably, the results from mold cavities S1 to S4 exhibited a high degree of similarity. Similarly, analyses of the fiber orientation tensor in the cross-flow direction (A_22_) and thickness direction (A_33_) in Figure 8b,c, respectively, demonstrated almost identical outcomes across cavities S1 to S4. In summary, within the NGR of this particular one-by-four cavity system, the fiber orientation remained consistent across different mold cavities. Furthermore, the investigation extended to exploring fiber orientation variations in the EFR of distinct mold cavities simultaneously, disregarding flow–fiber coupling effects. The examination of the fiber orientation tensor in the flow direction (A_11_), cross-flow direction (A_22_), and thickness direction (A_33_) in the four mold cavities revealed nearly equivalent results. In essence, within the EFR area of this one-by-four cavity system, the fiber orientation exhibited uniformity across various mold cavities. In conclusion, irrespective of the consideration of flow–fiber coupling effects, the fiber orientation distribution performance in each mold cavity remains largely consistent in a geometrically symmetrical one-by-four cavity system.

### 4.3. Discover the Evidence of the Flow–Fiber Coupling Effect in a Single Cavity S1

#### 4.3.1. Sprue Pressure

Figure 9 illustrates the variation in the sprue pressure history curve with the inclusion of flow–fiber coupling. The findings indicate that the incorporation of flow–fiber coupling results in the melt flow exhibiting higher sprue pressure in contrast to scenarios where there is no flow–fiber coupling system.

#### 4.3.2. Theoretical Investigation into the Impact of the Overflow Area on the Flow–Fiber Coupling Effect

In the literature [29,30], it was noted that in the EFR, the core layer region exhibits superior characteristics when the flow–fiber coupling effect is taken into account compared to when it is not considered. Specifically, mold cavities S1 to S4 were found to have identical fiber orientation distributions. Consequently, the focus was directed toward the detailed examination of the single mold cavity S1 system. Analysis of the EFR at position H3 revealed that there was minimal disparity in fiber arrangement distribution along the flow direction between scenarios with and without the flow–fiber coupling effect in Figure 10a, suggesting a potentially inconspicuous impact of this coupling in that area. This observation contrasts with previous research findings where the flow–fiber coupling effect was distinctly noticeable at the H3 location of a long flat plate system (shown in Figure 1a). Subsequently, the investigation was extended to the downstream H4 location, revealing a substantial reduction in the fiber orientation distribution along the flow direction when the flow–fiber coupling effect was considered, as shown in Figure 10b. This reduction led to a notable decrease in the core layer region area compared to the system without the flow–fiber coupling effect. In Figure 10c, quantitative analysis illustrated that the core layer region area decreased to approximately 0.45 when the flow–fiber coupling effect was not considered, while it increased to 0.55 when this effect was taken into account. The findings suggest a discernible impact of flow–fiber coupling in the H4 region based on theoretical analysis.

### 4.4. The Relationship between Fiber Orientations and Geometrical Shrinkage

#### 4.4.1. Numerical Prediction of the Average Fiber Orientation along Flow Direction

In the NGR or EFR, each designated measuring location (B1 to B5, or H1 to H5) will be subdivided into 11 measuring nodes along the thickness axis, as illustrated in Figure 1f. Subsequently, following the computation of injection molding simulation and fiber orientation analysis, the fiber orientation tensors A_11_, A_22_, and A_33_ at this specific point can be derived. These fiber orientation tensors can then be graphed on the 11 measuring nodes along the thickness direction to generate Figure 6, Figure 7, Figure 8 and Figure 9. To gain a deeper insight into the comparative fiber orientation distribution across various measuring locations, an approach involving the averaging of fiber orientation tensors for each measuring location is employed. The specific methodology for this process is outlined below. In this discussion, we will explore the correlation between microstructure (fiber orientations) and macroproperties (geometrical shrinkage). Initially, the fiber orientation distribution at various measurement points is analyzed using a method for integration [30]. Specifically, the fiber orientation tensors measured at each measurement point can be averaged as the following equation. The fiber orientation tensors, averaged along the flow direction from H1 to H5 in EFR, are shown in Figure 11.
(17)$${\left(A_{ii}\right)}_{avg}=\sum_{k=1}^{11}{\left(A_{ii}\right)}_{k}\;\;\;\;\;i=1\;\mathrm{to}\;3,$$
where k represents eleven points along the thickness direction at each measuring point.

Subsequently, the average values of A_11_, A_22_, and A_33_ are determined at each location. The outcomes are depicted in Figure 12. As illustrated in Figure 12a, within the NGR spanning from upstream B1 to downstream B5, there is a gradual increase in fiber orientation in the flow direction (A_11_), while the orientations in the cross-flow direction (A_22_) and thickness direction (A_33_) diminish progressively. Overall, the dominance lies with the orientation in the flow direction. Conversely, Figure 12b demonstrates that in the EFR, from upstream H1 to downstream H5, the analysis reveals a prevalence of A_11_ from H1 to H4. However, a notable shift occurs from H4 to H5, where A_11_ diminishes rapidly, and A_22_ and A_33_ experience a swift increase. This transition is primarily attributed to the flow–fiber coupling effect. In addition, compared with the results of previous research [30], as listed in Figure 12c, it is found that in the absence of an overflow area, the flow–fiber coupling effect will guide the fiber orientation in the flow direction (A_11_) and gradually turn to the cross-flow direction (A_22_) and thickness direction (A_33_). These changes are more gradual.

#### 4.4.2. Numerical prediction of the geometrical shrinkage

Figure 13a illustrates the shrinkage deformation of the injection-molded part (S1) within the NGR along the *x*-axis (flow) direction. The shrinkage measurement amounts to 0.1 mm (approximately 0.54%). Additionally, Figure 13b represents the shrinkage deformation along the *y*-axis (cross-flow) direction, with the upstream shrinkage value of (Ly)_L_ at 0.095 mm (1%) being lower than (Ly)_L_ at 0.139 mm (1.45%). Similarly, in the *z*-axis (thickness) direction, the shrinkage value of (Lz)_L_ at 0.0076 mm (0.22%) upstream is less than (Lz)_R_ at 0.016 mm (0.46%), as depicted in Figure 13c. In terms of process characteristics, the upstream (Ly)_L_ and (Lz)_L_ are closer to the gate and can theoretically be more effectively packed, resulting in smaller shrinkage. Moreover, Figure 13d displays the shrinkage deformation in the EFR along the *x*-axis (flow) direction, with a shrinkage value of 0.105 mm (approximately 0.56%). Figure 13e represents the *y*-axis (cross-flow) direction shrinkage deformation, where the upstream shrinkage value of (Ly)_L_ at 0.135 mm (1.5%) is higher than (Ly)_R_ at 0.086 mm (0.9%). Similarly, in the *z*-axis (thickness) direction shrinkage deformation, the upstream shrinkage value of (Lz)_L_ at 0.021 mm (0.6%) is also greater than (Lz)_R_ at 0.0025 mm (0.07%), as shown in Figure 13f. It is observed that the shrinkage deformation characteristics of the injection-molded part in the EFR are diametrically opposite to those in the NGR. These outcomes cannot be elucidated solely by pressure packing/holding characteristics.

#### 4.4.3. Experimental Validation of the Geometrical Shrinkage

Figure 13 and Table 1 demonstrate the dimensional measurement of each side of the injected parts. Specifically, Figure 14a illustrates the observed shrinkage and deformation of the injected finished part (S1) in the NGR along the *x*-axis (flow) direction. The actual expansion measurement is 0.17 mm (0.9% expansion). Additionally, Figure 14b displays the shrinkage deformation in the *y*-axis (cross-flow) direction, with a shrinkage value of 0.10 mm (1.08%) for (Ly)_L_ upstream, which is less than 0.12 mm (1.22%) for (Ly)_L_. Similarly, in the *z*-axis (thickness) direction, the shrinkage deformation is shown in Figure 14c, where the shrinkage value of 0.02 mm (0.57%) for (Lz)_L_ upstream is less than the 0.03 mm (0.86%) for (Lz)_R_. The injected part’s process characteristics indicate that the upstream (Ly)_L_ and (Lz)_L_, being closer to the gate, can be more effectively packed, resulting in smaller shrinkage. The overall trend reveals consistent asymmetric shrinkage values in the cross-flow and thickness directions, as indicated by the simulation analysis obtained. Moreover, Figure 14d presents the actual shrinkage deformation in the EFR along the *x*-axis (flow) direction, with an actual expansion value of 0.12 mm (0.65% expansion). Figure 14e depicts the amount of shrinkage deformation in the *y*-axis (cross-flow) direction, where the shrinkage value of 0.18 mm (1.9%) for (Ly)_L_ upstream is greater than the 0.1 mm (1.1%) for (Ly)_R_ downstream. Similarly, in the *z*-axis (thickness) direction, the shrinkage deformation is shown in Figure 14f, with the shrinkage value of 0.063 mm (1.8%) for (Lz)_L_ upstream being greater than the 0.013 mm (0.38%) for (Lz)_R_ downstream. The shrinkage deformation characteristics of the injected part in the EFR are found to be entirely opposite to those in the NGR. The overall trend aligns well with the simulation analysis, particularly regarding the asymmetric shrinkage characteristics in the cross-flow and thickness directions.

## 5. Discussion

### 5.1. Flow–Fiber Coupling Effect Validation

In this study, a 1 × 4 multi-cavity system with a geometrically symmetrical design was employed. This design differs significantly from the previously utilized large single mold cavity geometry (see Ref. [30]) or circular plate (see Ref. [22]). Two key observations were made: (1) the sprue pressure in the system with flow–fiber coupling exceeded that in the system lacking this coupling, and (2) the core layer region in the system with flow–fiber coupling was found to be expanded in comparison to the system without such coupling. These findings align well with the existing literature on the subject.

### 5.2. The Influence of Overflow Area on the Occurrence of Flow–Fiber Coupling

The time of occurrence to obviously observe the flow–fiber coupling is contrasted with findings from prior research [29,30]. The analysis reveals that in systems without overflow areas, such as the plate system depicted in Figure 1a, the timing of fiber coupling with the flow is typically evident at the H3 location, as shown in Figure 15. However, the incorporation of an overflow area in this investigation altered the flow dynamics, subsequently influencing fiber orientation. These modifications in fiber orientation then interacted with the flow of molten material. Notably, the time of occurrence of obvious flow–fiber coupling was not distinctly discernible until the H4 location, as illustrated in Figure 10b. Additionally, the timing of flow–fiber coupling is also depicted in Figure 12b. Consequently, the presence of an overflow area prolongs the manifestation of the discernible flow–fiber coupling effect. In brief, due to the presence of a small bridge measuring 7.0 mm × 5.1 mm × 0.9 mm and the extra circle domain in the overflow regions, the flow characteristics of the polymer melt are altered. This component extends the flow path in the direction of flow, enabling the flow field to exert greater influence over the flow dynamics. Consequently, the discharge direction A11 is obstructed towards the end of the filling process, leading to a delayed manifestation of the flow–fiber coupling effect.

### 5.3. Validation of the Flow–Fiber Coupling Effect on the Asymmetrical Shrikange Behavior from Upstream to Downstream

The shrinkage behavior of the injected product in the EFR and NGR exhibits contrasting characteristics, as evident from the simulation results in Figure 13 and experimental observations in Figure 14. The asymmetrical shrinkage phenomenon, where the cross-flow direction (Ly)_L_ and thickness direction (Lz)_L_ exhibit greater shrinkage compared to (Ly)_R_ and (Lz)_R_, respectively, resulting in a larger geometry at the upstream end than at the downstream end, cannot be solely explained by the packing/holding pressure properties of melt. Upon closer examination of Figure 12b and Figure 13e,f, particularly in the downstream region from H4 to H5, a notable shift in fiber arrangement from the flow direction (A_11_) to the cross-flow direction (A_22_) and thickness direction (A_33_) is observed. The accumulation of fibers in the downstream H4 to H5 area, oriented in the cross-flow direction (A_22_) and thickness direction (A_33_), effectively resists the shrinkage of the injected product in this region. Consequently, the asymmetrical fiber orientations in the cross-flow and thickness directions between the upstream (H1 to H2) and downstream (H4 to H5) areas significantly influence the shrinkage behavior, surpassing the conventional packing/holding pressure effects experienced by typical materials and resulting in greater upstream shrinkage compared to downstream shrinkage. These geometric alterations align well with the evolving trend of the fiber orientation effect derived from theoretical analysis.

## 6. Conclusions

According to the findings from the study on the flow–fiber coupling effect of a 1 × 4 multi-cavity system featuring a geometrically symmetrical configuration, the subsequent conclusions can be derived.

Irrespective of the consideration of the flow–fiber coupling effect, the theoretical fiber orientation of each cavity remains consistent.When examining the flow–fiber coupling effect, two observations can demonstrate this phenomenon: (1) the sprue pressure in the system exhibiting flow–fiber coupling was higher than in the system without this coupling, and (2) the core layer area in the system with flow–fiber coupling was observed to be enlarged compared to the system lacking such coupling.In a geometric system featuring an overflow area design, the presence of these overflow areas serves to postpone the onset of the flow–fiber coupling effect in contrast to a system lacking such design elements.Through the integration of fiber orientation distribution data, it is possible to examine the consequences of the delayed flow–fiber coupling effect in the overflow area, particularly in the EFR. This phenomenon leads to a significant transfer of A_11_ fiber orientation to A_22_ and A_33_ in the final filling zone (H4 to H5), resulting in an asymmetrical arrangement of fiber orientation.By analyzing the geometric dimensions of the final part from the 1 × 4 multi-cavity system with overflow area, it is evident that there are noticeable variations in size and shrinkage between the upstream and downstream sections within the EFR. These differences align with the asymmetrical distribution of fiber orientation induced by the flow–fiber coupling effect.

## Figures and Tables

**Figure 1 polymers-16-01279-f001:**
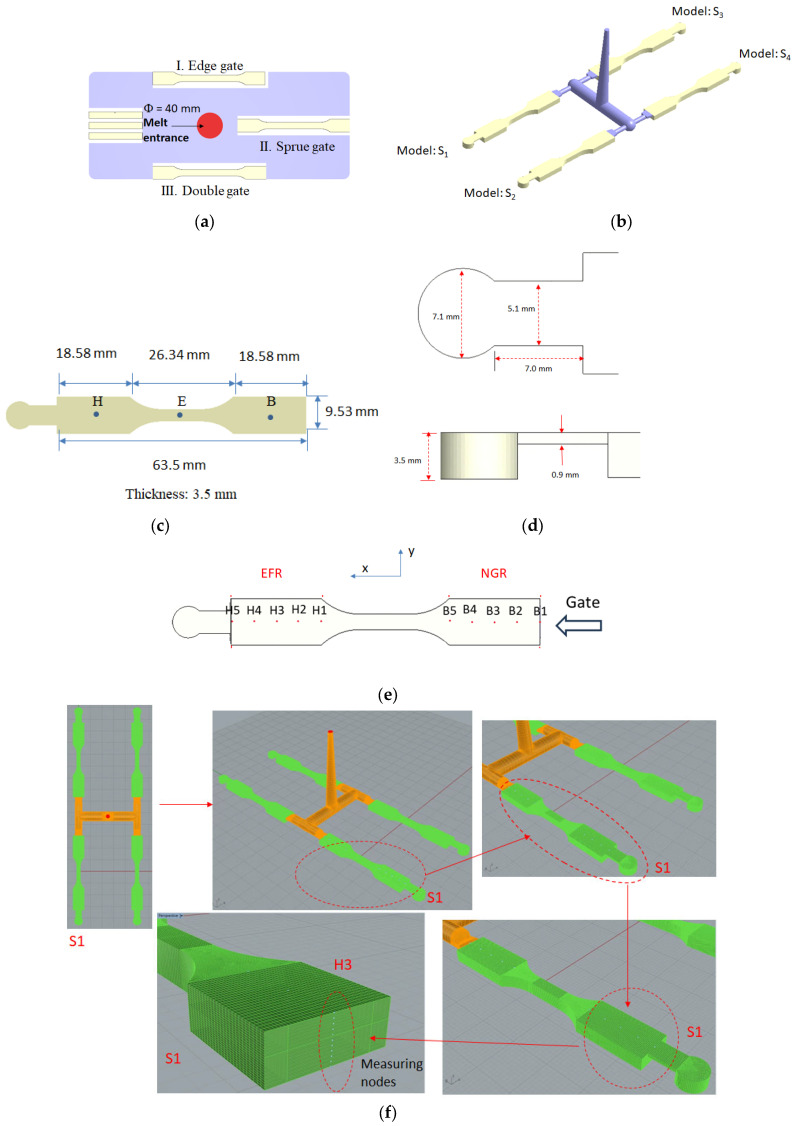
(**a**) the geometric configuration adapted from the previous study [30]; (**b**) the geometric configuration in this study; (**c**) the dimension of Models S1 to S4; (**d**) the structure and dimension of the overflow area; (**e**) the definition of the measuring locations for each model. Here, the variable x corresponds to the flow direction, while y pertains to the cross-flow direction; (**f**) Procedure for setting measuring nodes in the thickness direction to capture the associated fiber orientation data.

**Figure 2 polymers-16-01279-f002:**
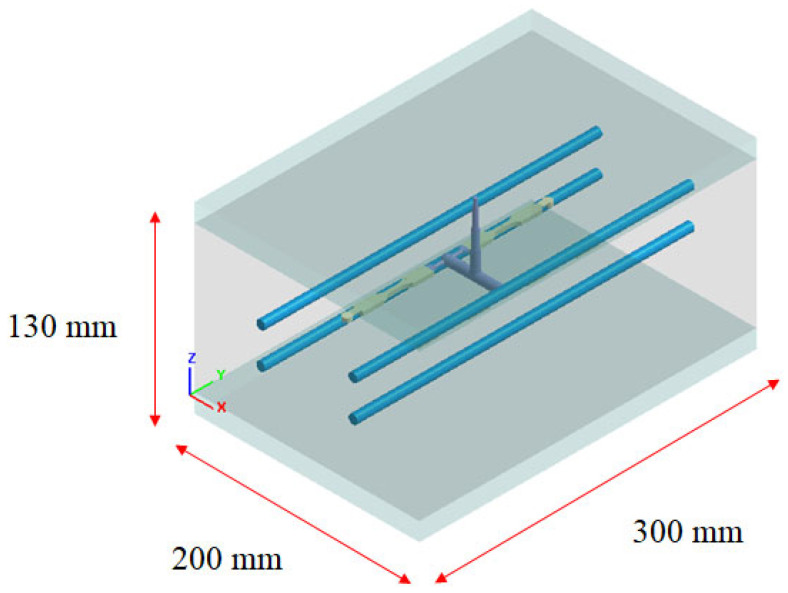
The configuration of the mold base and the arrangement of cooling channels.

**Figure 3 polymers-16-01279-f003:**
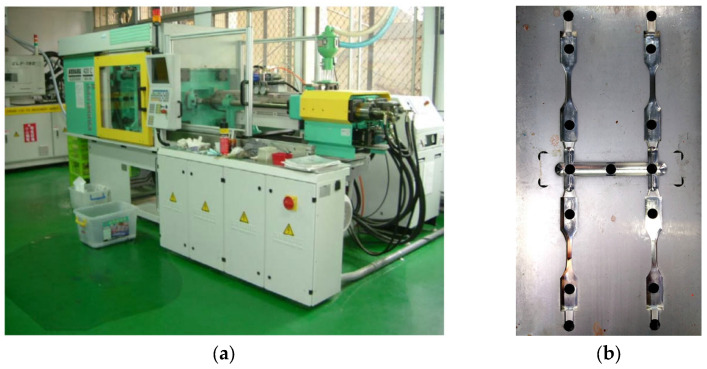
(**a**) The injection machine: Arburg 420C 1000-350-40; (**b**) the cavity structure.

**Figure 4 polymers-16-01279-f004:**
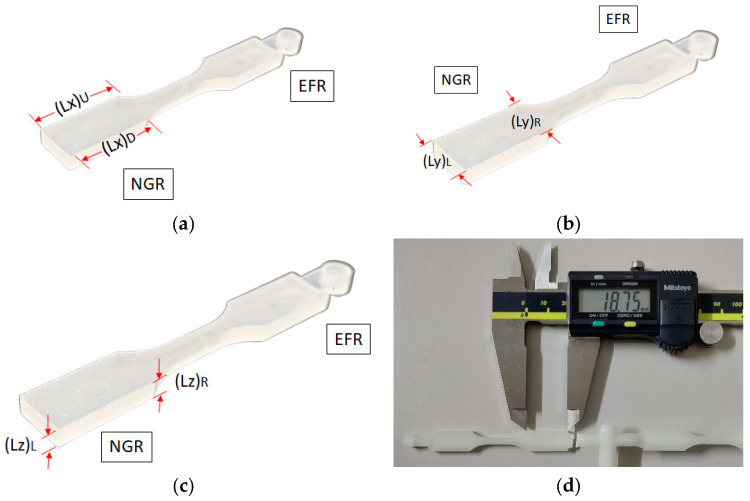
Definition of the side length code of the injected parts (taking NGR as an example): (**a**) the two sides in the flow direction (Lx)_U_ and (Lx)_D_; (**b**) the two sides in the cross-flow direction (Ly)_L_ and (Ly)_R_; (**c**) the two sides in the thickness direction (Lz)_L_ and (Lz)_R_; (**d**) using a vernier ruler to measure the length of each side.

**Figure 5 polymers-16-01279-f005:**
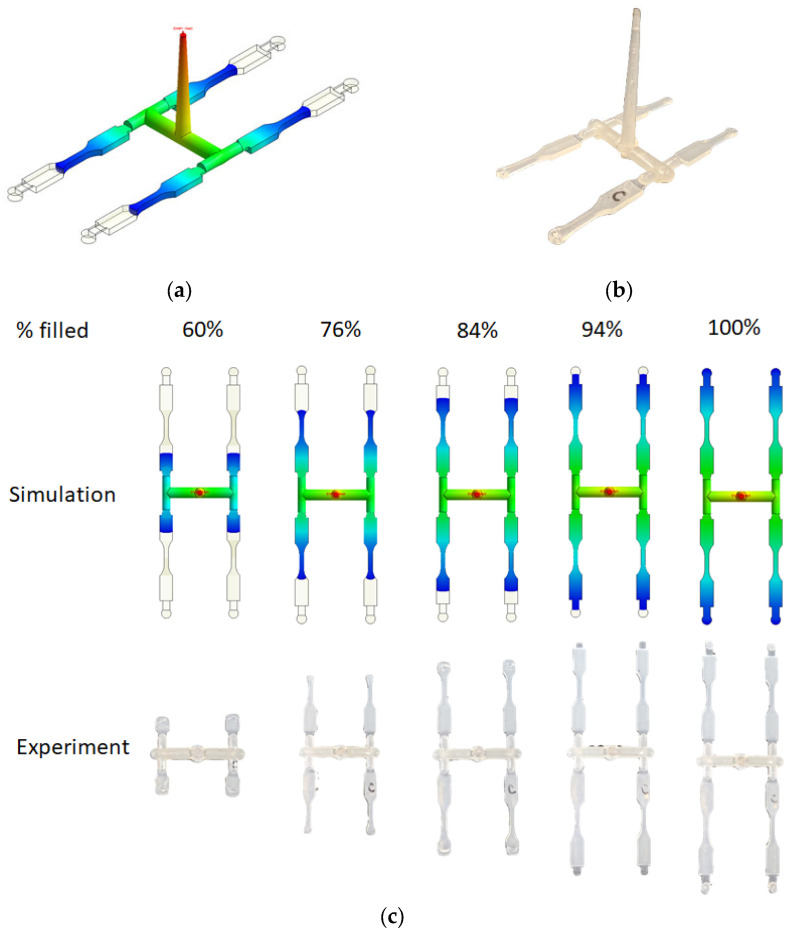
(**a**) Melt flow front behavior at 76% volume filled by CAE simulation; (**b**) melt flow front behavior at 76% volume filled by experiment; (**c**) the comparison of the melt flow front between simulation and experiment from 60% to 100% volume filled.

**Figure 6 polymers-16-01279-f006:**
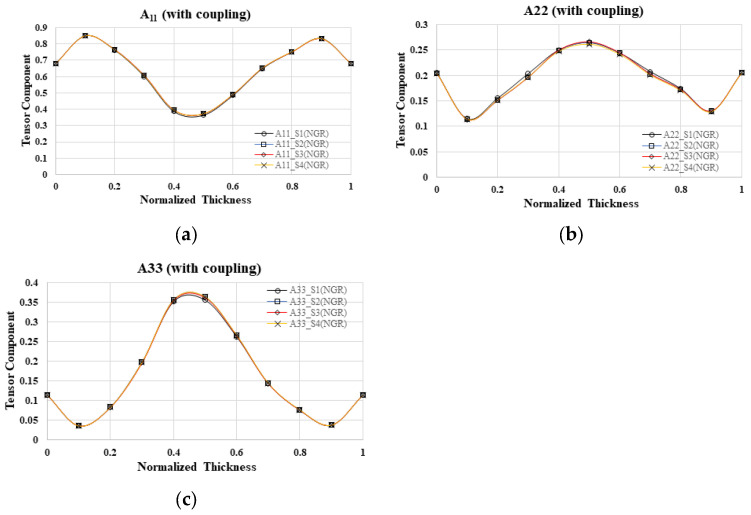
Fiber orientation distribution with flow–fiber coupling at B3 of NGR, where A_11_ is fiber orientation tensor in flow direction (**a**); A_22_ is fiber orientation tensor in cross-flow direction (**b**); A_33_ is fiber orientation tensor in thickness direction; S1 to S4 is the cavity number; NGR is the near gate region (**c**).

**Figure 7 polymers-16-01279-f007:**
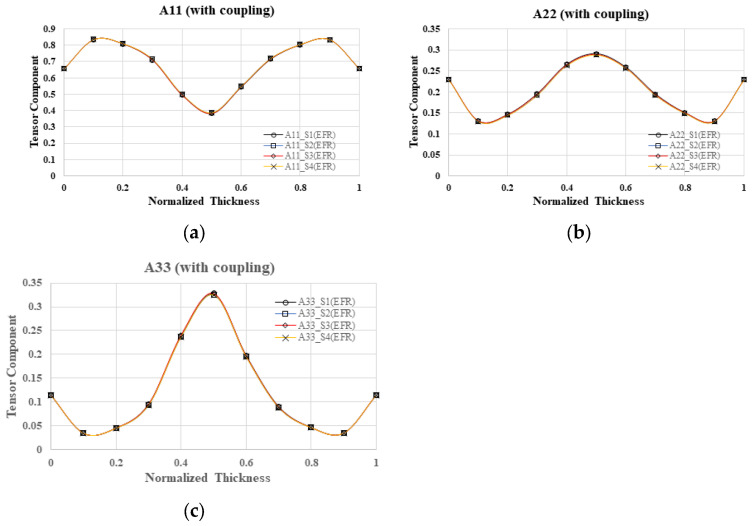
Fiber orientation distribution with flow–fiber coupling at H3 of the EFR, where A_11_ is fiber orientation tensor in flow direction (**a**); A_22_ is fiber orientation tensor in cross-flow direction; A_33_ is fiber orientation tensor in thickness direction (**b**); S1 to S4 is the cavity number; EFR is the end of filling region (**c**).

**Figure 8 polymers-16-01279-f008:**
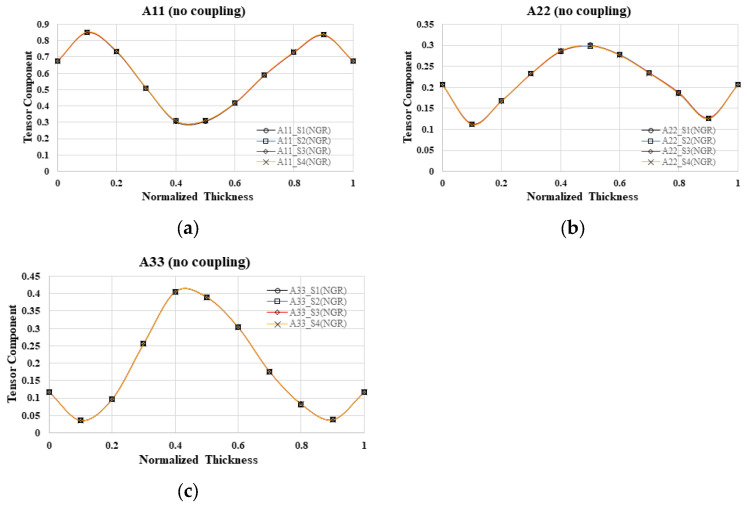
Fiber orientation distribution without flow–fiber coupling at B3 of NGR, where A_11_ is fiber orientation tensor in flow direction (**a**); A_22_ is fiber orientation tensor in cross-flow direction; A_33_ is fiber orientation tensor in thickness direction (**b**); S1 to S4 is the cavity number; NGR is the near gate region (**c**).

**Figure 9 polymers-16-01279-f009:**
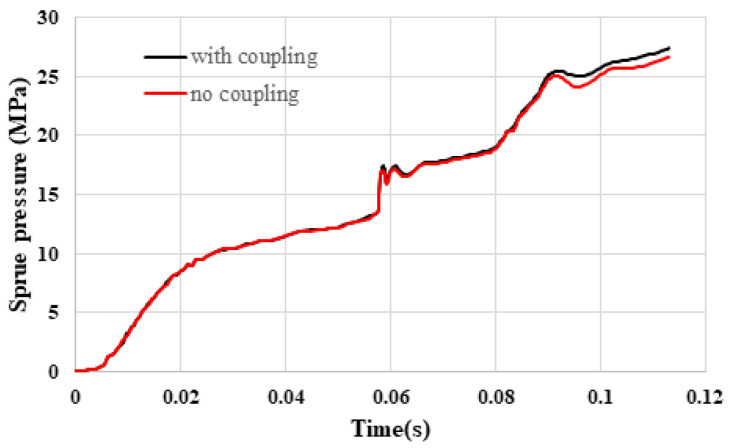
Sprue pressure history curve of the injection system with or without flow–fiber coupling.

**Figure 10 polymers-16-01279-f010:**
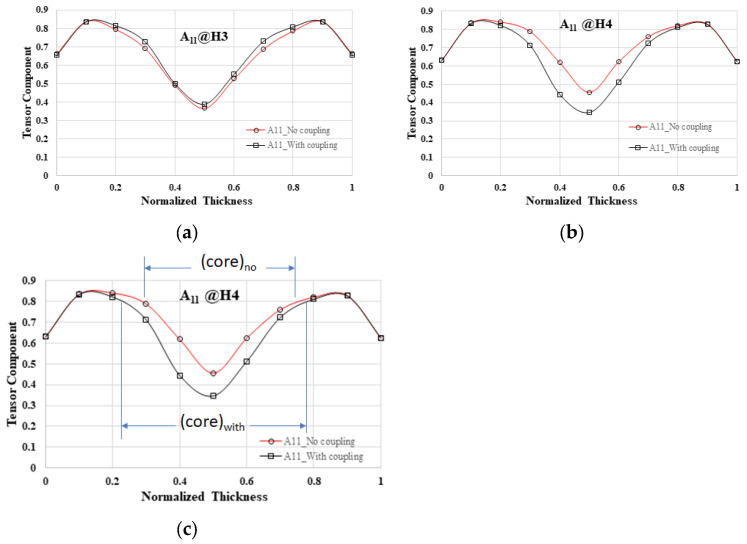
The comparison of fiber orientation distribution between systems with and without flow–fiber coupling in the system with overflow area: (**a**) at H3 of the EFR; (**b**) at H4 of the EFR; (**c**) the comparison of the core layer region between systems with and without flow–fiber coupling.

**Figure 11 polymers-16-01279-f011:**
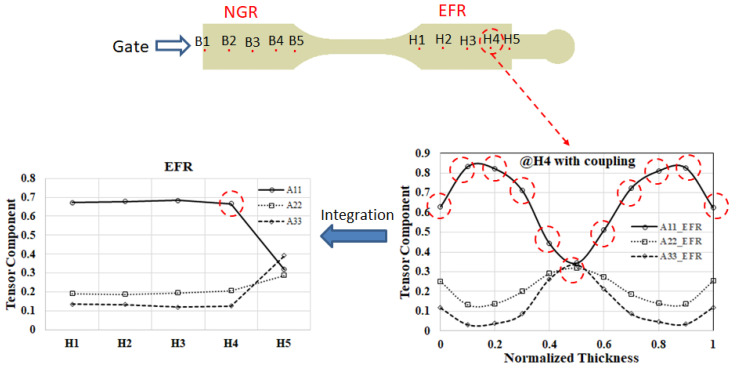
The averaged fiber orientation tensor at each location can be obtained along the flow direction.

**Figure 12 polymers-16-01279-f012:**
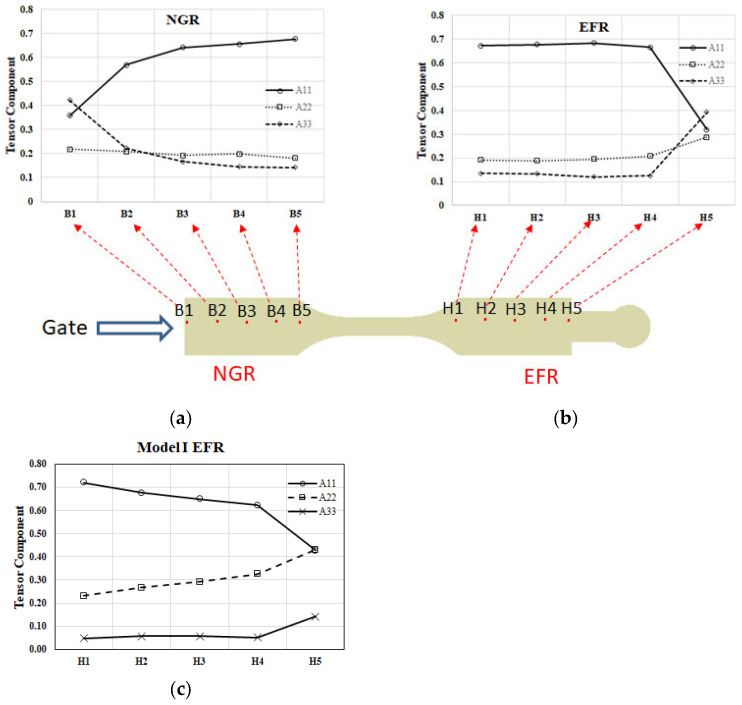
The average fiber orientation tensors along the flow direction in the system with overflow area: (**a**) at NGR; (**b**) at EFR; (**c**) Model I at EFR of a plate system Adapted from [30].

**Figure 13 polymers-16-01279-f013:**
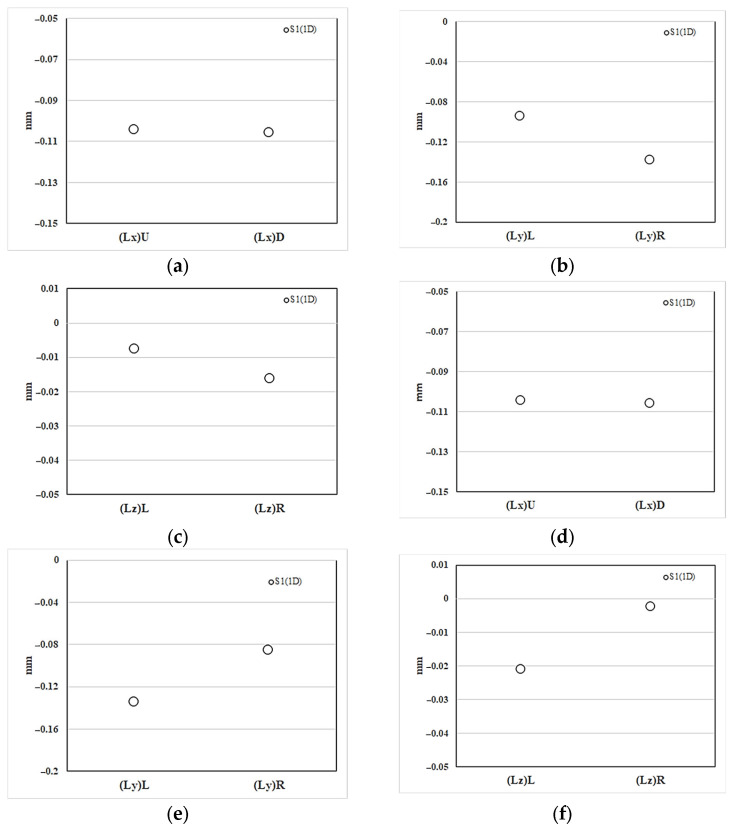
The numerical prediction of the geometrical shrinkage along the flow direction in the system with overflow area: (**a**–**c**) at NGR; (**d**–**f**) at EFR.

**Figure 14 polymers-16-01279-f014:**
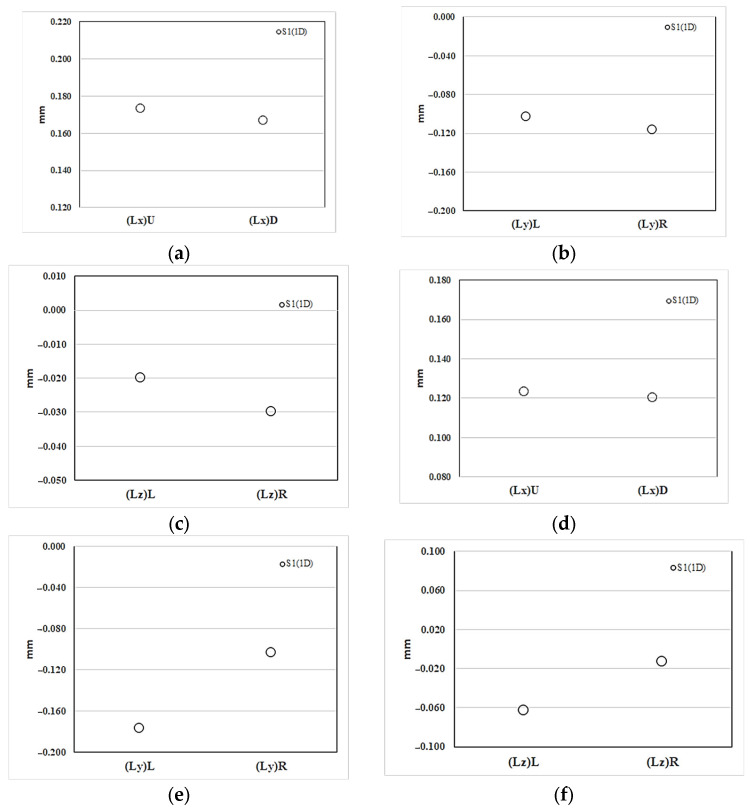
The experimental measurement of the geometrical shrinkage along the flow direction in the system with overflow area: (**a**–**c**) at NGR; (**d**–**f**) at EFR.

**Figure 15 polymers-16-01279-f015:**
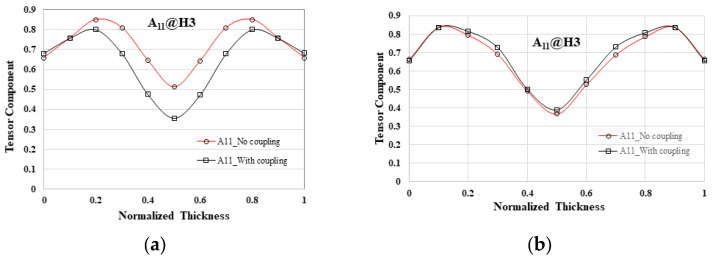
The comparison of the occurrence of flow–fiber coupling effect at H3 of EFR for different geometrical systems: (**a**) a plate system as shown in Figure 1a [30]; (**b**) in this study.

**Table 1 polymers-16-01279-t001:** The experimental measurement of the dimensions of the individual sides of the injected parts.

Region	Material	Shrinkage						
		Unit	(Lx)_U_	(Lx)_D_	(Ly)_L_	(Ly)_R_	(Lz)_L_	(Lz)_R_
EFR	PP	mm	18.440	18.440	9.317	9.333	3.440	3.450
			±0.000	±0.000	±0.006	±0.006	±0.010	±0.010
		%	−0.753	−0.753	−2.239	−2.064	−1.714	−1.429
	30SFPP	mm	18.703	18.700	9.353	9.427	3.437	3.487
			±0.006	±0.000	±0.006	±0.006	±0.006	±0.006
		%	0.664	0.646	−1.854	−1.084	−1.810	−0.381

## Data Availability

The data presented in this study are available on request from the corresponding author.

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
