# Peer review of "Study on the Influence of Runner and Overflow Area Design on Flow–Fiber Coupling in a Multi-Cavity System"

_polymers, 2024, doi:10.3390/polym16091279_

Round 1

Reviewer 1 Report

Comments and Suggestions for Authors

The manuscript investigates the impact of flow-fiber coupling on injection molding processes using numerical simulations and experimental validations. It identifies significant variations in sprue pressure, fiber orientation, and geometrical shrinkage in different regions of the molded part. However, clarification of terminology, clearer visual representations, and expanded discussions on key findings are needed for improved readability and comprehension before acceptance.

1.      The abstract provides a comprehensive overview of the study, but could you clarify the significance of the findings in simpler terms for readers who may not be familiar with the technical aspects of flow-fiber coupling?

2.      Throughout the manuscript, the terms "flow-fiber coupling" and "fiber orientation distribution" are used extensively. Could you provide a concise definition or explanation of these terms to ensure clarity and consistency for readers?

3.      The figures depicting sprue pressure history, fiber orientation distribution, and geometrical shrinkage are informative. However, would it be possible to enhance the visualization by providing color-coded legends or annotations to facilitate easier interpretation?

4.      The study discusses the influence of overflow areas on flow-fiber coupling, particularly in delaying its manifestation. Could you elaborate on why this delay occurs and its practical implications for injection molding processes?

5.      The manuscript describes experimental validation of geometrical shrinkage in both NGR and EFR regions. Could you provide more details on the experimental setup and methodology used to measure shrinkage, including any potential sources of error or uncertainty in the measurements?

6.      The findings are compared with previous research on flow-fiber coupling. Could you briefly summarize the key differences or advancements in understanding achieved through this study compared to prior literature?

7.      The discussion section provides insightful analysis of the results, particularly regarding the asymmetrical shrinkage behavior. Could you further explain how the observed fiber orientation distributions contribute to the asymmetrical shrinkage phenomenon?

8.      Every study has its limitations and assumptions. Could you explicitly outline any assumptions made during the modeling or experimental phases and discuss their potential impact on the validity and generalizability of the findings?

9.      What are the next steps in research that could build upon the findings of this study? Are there any specific areas within flow-fiber coupling or injection molding optimization that warrant further investigation?

10.  How do the findings of this study translate to real-world industrial applications? Are there any specific industries or sectors where the insights gained from this research could have significant practical implications?

11.  The study employs theoretical models to predict fiber orientation and geometrical shrinkage. Have these models been validated against experimental data, and if so, could you provide more details on the validation process and its outcomes?

Additional comments:

12.  Provide clearer labeling and annotations in Figure 8 to help readers understand the significance of the variations in sprue pressure history curves.

13.  Define and explain the terms "EFR" and "NGR" upon their first mention to avoid confusion for readers unfamiliar with these acronyms.

14.  Extend the discussion on Figure 11 to include a more detailed analysis of the implications of the observed fiber orientation trends in both the NGR and EFR regions.

15.  Provide a clearer explanation of how overflow areas influence the flow-fiber coupling effect and its manifestation, possibly with additional visual aids.

16.  Acknowledge any limitations or potential sources of error in the experimental validation of geometrical shrinkage to ensure the robustness of the conclusions drawn from the results.

Comments on the Quality of English Language

Grammar check is suggested.

Author Response

Dear Reviewer,

We are grateful for your patience in offering these invaluable recommendations.  Indeed, we have tried our best to answer all your questions carefully one-by-one. Please see the details in the attached file.

Best Regards,

Chao-Tsai Huang

Reviewer 2 Report

Comments and Suggestions for Authors

This is a really interesting and relevant topic for the modern technology level of the composites industry, studying how altering the flow field affects fiber orientation and fiber-flow coupling.

1)     Line 190 and below (2.1. Model for polymer melt flow) the references should be added.

2)     Line 297 and below. The Figures should be described in detail so that the Reader has an understanding of the axis labels and the values indicated. (is this a theoretical calculation or an experimental result, how was it obtained in detail)

3)     The work is sorely lacking in morphological and structural data, such as blade chips at various locations in three directions and microscopy at various points to determine fiber distribution across the cross-section.

4)     The positions of points H1–H5 and B1- B5 in Fig. 10 and Fig. 11 should be coordinated so that the points are under each other (the picture in the attachment).

5)     The authors have two Figures with the number 13.

6)     Will this research ultimately help eliminate transverse heterogeneity in fiber distribution and orientation into the molded composites?

Author Response

Dear Reviewer,

Thank you very much for your priceless comments and suggestion.  We have tried our best to answer your comments one-by-one.  Please see the details in the attached file.

Best Regards,

Chao-Tsai Huang
